# Exploring the Mechanism of Flaccidoxide-13-Acetate in Suppressing Cell Metastasis of Hepatocellular Carcinoma

**DOI:** 10.3390/md18060314

**Published:** 2020-06-15

**Authors:** Yu-Jen Wu, Wen-Chi Wei, Guo-Fong Dai, Jui-Hsin Su, Yu-Hwei Tseng, Tsung-Chang Tsai

**Affiliations:** 1Department of Beauty Science, Meiho University, Pingtung 91202, Taiwan; x00002180@meiho.edu.tw; 2Department of Food and Nutrition, Meiho University, Pingtung 91202, Taiwan; 3Yu Jun Biotechnology Co., Ltd., Kaohsiung 807, Taiwan; fwind101@gmail.com; 4National Research Institute of Chinese Medicine, Taipei 112, Taiwan; jackwei@nricm.edu.tw (W.-C.W.); mayeeshat@nricm.edu.tw (Y.-H.T.); 5National Museum of Marine Biology and Aquarium, Pingtung 94450, Taiwan; x2219@nmmba.gov.tw; 6Antai Medical Care Corporation Antai Tian-Sheng Memorial Hospital, Pingtung 92842, Taiwan

**Keywords:** flaccidoxide-13-acetate, hepatocellular carcinoma, invasion, migration, epithelial-mesenchymal transition

## Abstract

Hepatocellular carcinoma (HCC) is the most common liver or hepatic cancer, accounting for 80% of all cases. The majority of this cancer mortality is due to metastases, rather than orthotopic tumors. Therefore, the inhibition of tumor metastasis is widely recognized as the key strategy for successful intervention. A cembrane-type diterpene, flaccidoxide-13-acetate, isolated from marine soft coral *Sinularia gibberosa*, has been reported to have inhibitory effects against RT4 and T24 human bladder cancer invasion and cell migration. In this study, we investigated its suppression effects on tumor growth and metastasis of human HCC, conducting Boyden chamber and Transwell assays using HA22T and HepG2 human HCC cell lines to evaluate invasion and cell migration. We utilized gelatin zymography to determine the enzyme activities of matrix metalloproteinases MMP-2 and MMP-9. We also analyzed the expression levels of MMP-2 and MMP-9. Additionally, assays of tissue inhibitors of metalloproteinase-1/2 (TIMP-1/2), the focal adhesion kinase (FAK)/phosphatidylinositide-3 kinases (PI3K)/Akt/mammalian target of the rapamycin (mTOR) signaling pathway, and the epithelial-mesenchymal transition (EMT) process were performed. We observed that flaccidoxide-13-acetate could potentially inhibit HCC cell migration and invasion. We postulated that, by inhibiting the FAK/PI3K/Akt/mTOR signaling pathway, MMP-2 and MMP-9 expressions were suppressed, resulting in HCC cell metastasis. Flaccidoxide-13-acetate was found to inhibit EMT in HA22T and HepG2 HCC cells. Our study results suggested the potential of flaccidoxide-13-acetate as a chemotherapeutic candidate; however, its clinical application for the management of HCC in humans requires further research.

## 1. Introduction

Among the primary liver cancers, hepatocellular carcinoma (HCC) is especially common, and the majority of liver cancer deaths can be attributed to HCC [1]. Worldwide, it ranks sixth in terms of cancer incidence and is one of the deadliest types of cancer [2]. Countries in Asia and sub-Saharan Africa, in particular, suffer severely from this epidemic, patients in China, North and South Korea, Vietnam, and sub-Saharan African countries accounting for almost 82% of liver cancer cases [3]. Hepatitis B and C viruses are the main etiological factors in the Asia-Pacific region, chronic hepatitis B (CHB) infections being the major cause of liver cancer [3,4]. In addition, antiviral medication for CHB and hepatitis C (CHC) is a proven treatment by which to reduce the incidence of HCC [5]. Currently, HCC treatment options include surgical resection, radiotherapy, trans-arterial chemoembolization, radioembolization, targeted therapy, and liver transplantation, depending on the tumor progression [6,7]. Recent advances in surgical treatments and loco regional therapies have significantly improved the short-term survival of HCC patients [8]. However, the recurrence or metastasis of HCC remain serious concerns, representing an unmet need that warrants the development of new drugs for patients with malignant HCC. Marine organisms offer one of the richest sources of leading candidates for drug development due to the highly diversified marine environment [9]. Since the 1960s, the isolation and identification of many novel compounds have been performed. Marine-based drugs to combat cancer and other diseases have been developed and introduced in clinical settings [10,11]. Existing in warm seawater bodies, soft corals possess unique and abundant secondary metabolites that have been investigated with regards to various biological activities, such as anti-inflammatory and anticancer effects [12,13]. 

A cembrane-type diterpenoid, flaccidoxide-13-acetate, isolated from marine soft coral *Sinularia gibberosa*, has been suggested to induce apoptosis in human bladder cancer cells [14]. To our knowledge, its effect on HCC remains to be confirmed. This study aimed to fill the gap by investigating the inhibitory effects of this compound on the tumor growth and metastasis of human HCC. 

## 2. Results

### 2.1. Cytotoxic Activity of Flaccidoxide-13-acetate against HA22T and HepG2 Cells

We used a methylthiazole tetrazolium (MTT) assay to measure the cytotoxic effects of flaccidoxide-13-acetate at different concentrations (0, 2, 4, 6, and 8 μM) on human HCC cell lines HA22T and HepG2, the results of which are shown in Figure 1. Flaccidoxide-13-acetate was found to inhibit the cell viability of HA22T and HepG2 cells in a dose-dependent manner (* *p* < 0.01 and # *p* < 0.05) (Figure 1). Only at 8 μM did HA22T and HepG2 cells exhibit a cell viability below 80% (Figure 1). Therefore, we selected lower concentrations (2–8 μM) for uses in all subsequent experiments.

### 2.2. Flaccidoxide-13-acetate Reduces Cellular Migration and Invasion in HA22T and HepG2 Cells 

The cell–matrix interaction and cell motility play crucial roles in determining the metastatic properties of cancer cells. To investigate the inhibitory properties of flaccidoxide-13-acetate on the tumor growth and metastasis of human HCC, we conducted Boyden and Transwell chamber assays in HA22T and HepG2 human HCC cell lines to evaluate cell migration and invasion. Flaccidoxide-13-acetate was found to inhibit the cell migration of HA22T and HepG2 cells in a dose-dependent manner. After treatments of HA22T and HepG2 cells with 8-μM flaccidoxide-13-acetate for 24 h, the cell migratory capacities dropped to 85% and 80%, respectively (Figure 2). Similar reductions were observed in the invasive capacities of both cell lines (Figure 3).

### 2.3. Flaccidoxide-13-acetate Reduced the uPA and MMP-2/-9 Activities and Regulated the Expressions of MMP-2/-9, uPA, TIMP-1, and TIMP-2 in HA22T and HepG2 Cells

To investigate the enzymatic activities of MMP-2, MMP-9, and uPA produced from HA22T and HepG2 cells, gelatin zymography was used. We cultured HA22T and HepG2 cells in a serum-free media with flaccidoxide-13-acetate (0, 2, 4, 6, and 8 μM) for 24 h and collected the conditioned media to determine the activities of MMP-2, MMP-9, and the urokinase plasminogen activator (uPA). Flaccidoxide-13-acetate was found to inhibit the activities of MMP-2, MMP-9, and uPA, exhibiting a dose-dependent effect, as shown in Figure 4A. To clarify the roles of the endogenous tissue inhibitors of metalloproteinases (TIMPs) and serine protease uPA in regulating the activities of MMP-2, MMP-9, and MMP-13 after treatments with flaccidoxide-13-acetate, we performed Western blotting to measure the extent to which flaccidoxide-13-acetate regulated the expressions of the MMP-2, MMP-9, MMP-13, uPA, TIMP-1, and TIMP-2 proteins. We observed declines in the protein levels of MMP-2, MMP-9, MMP-13, and uPA, while, on the contrary, the expressions of TIMP-1 and TIMP-2 increased in HA22T and HepG2 cells after treatments with flaccidoxide-13-acetate for 24 h (Figure 4B).

### 2.4. Flaccidoxide-13-acetate Inhibited the FAK/PI3K/Akt/mTOR Signaling Pathway

Focal adhesion kinase (FAK) is cytoplasmic protein kinase in the cytoplasm. The overactivation of FAK has been recognized to promote the progression and metastasis of tumors via the regulation of cell motility, survival, and proliferation. To investigate the inhibitory effects of flaccidoxide-13-acetate on the FAK activity and downstream signaling pathways in HA22T and HepG2 cells, we adopted the Western blotting technique to measure the extent to which flaccidoxide-13-acetate regulated the expressions of FAK, PI3K, AKT, mTOR, p-PI3K, mTOR, and p-mTOR proteins. Declines in the protein levels of FAK, p-PI3K, mTOR, and p-mTOR were observed (Figure 5).

### 2.5. Flaccidoxide-13-acetate Inhibited the Epithelial-to-Mesenchymal Transition (EMT)

EMT is a process in which epithelial cells lose adherence and become mesenchymal cells, playing a significant role in tumor metastasis. During EMT, Snail, a key transcription factor for EMT, inhibits E-cadherin transcription, and vimentin regulates β-catenin translocation. 

In order to investigate whether flaccidoxide-13-acetate inhibited EMT in HA22T and HepG2 cells, Western blotting was used to measure the expressions of cytosolic β-catenin, N-cadherin, E-cadherin, vimentin, and nucleic Snail. The results showed increases in the protein levels of N- and E-cadherin, while, on the contrary, those of β-catenin, vimentin, and Snail decreased in HA22T and HepG2 cells after treatments with flaccidoxide-13-acetate (Figure 6).

## 3. Discussion

Metastases contribute to the majority of cancer deaths, rather than orthotopic tumors. Therefore, the inhibition of tumor metastasis is generally recognized as a key approach for intervention [15,16]. Previous studies of human bladder cancer cells showed that flaccidoxide-13-acetate reduced apoptosis, which is mediated by p38/JNK activation, mitochondrial dysfunction, and endoplasmic reticulum stress [14]; it was also found to inhibit human bladder cancer cell migration and invasion by reduction in the activation of the FAK/PI3K/AKT/mTOR signaling pathway [17]. In this study, we investigated the therapeutic effects of flaccidoxide-13-acetate on tumor metastasis in human HCC. A HA22T cell line was established from a specimen obtained from a 56-year-old Chinese male HCC patient [18], while a HepG2 cell line was obtained from an HCC sample from a 15-year-old Caucasian male [19]. We analyzed the therapeutic effects of flaccidoxide-13-acetate on tumor metastasis in HA22T and HepG2 cell lines, and thereafter, investigated its possible mechanisms of action. Our results showed that, at certain concentrations, flaccidoxide-13-acetate inhibited metastasis in HA22T and HepG2 cells. Reports published in the literature indicated that MMP-2 can enhance the migration of HCC cells by increasing fibronectin cleavage [20] and that the overexpression of MMP-9 is associated with HCC tumor migration and invasion [21]. 

MMP-2 and MMP-9 have been found to be involved in the metastasis, invasiveness, and prognosis of HCC. Flaccidoxide-13-acetate inhibited the activation of MMP-2, MMP-9, and uPA, whose physiological activities are significantly correlated with TIMPs and uPA. Research has also suggested that the expression of uPA, a key molecule contributing to the activation of MMPs, is associated with cancer invasion [21,22,23,24]. The results of the present study showed that flaccidoxide-13-acetate upregulated proteins TIMP-1 and TIMP-2 and downregulated the expression of the uPA protein in treated HCC cells. We suggest that flaccidoxide-13-acetate upregulates the TIMP-1 and TIMP-2 protein expressions and downregulates the uPA protein expression, resulting in the suppression of MMP-2 and MMP-9 activities in HA22T and HepG2 cells.

Focal adhesion kinase (FAK) plays significant roles in cell survival, motility, and proliferation and is associated with EMT, cancer invasion, and migration [25,26,27]. FAK has been reported to be associated with PI3K/Akt/mTOR signal transduction [28]. The PI3K/Akt/mTOR signaling pathway is also associated with EMT, cancer invasion, and migration. In this study, we showed that flaccidoxide-13-acetate inhibited FAK protein expression and the phosphorylation of PI3K, Akt, and mTOR. We suggest that flaccidoxide-13-acetate may contribute to the inhibition of FAK and the subsequent suppression of PI3K/Akt/mTOR signaling, interrupting the EMT and the invasion and migration of HCC cells.

Snail facilitates the initiation of the EMT and has been found to downregulate the transcription of E-cadherin, which is known to be important in cell–cell adhesion [29,30]. Therefore, Snail and E-cadherin are key markers of EMT [31,32,33]. Our data showed that flaccidoxide-13-acetate suppressed the protein expression of Snail in the nucleus and promoted the expression of E-cadherin in the cytosol. We suggest that flaccidoxide-13-acetate may suppress the EMT process, including the downregulation of Snail and upregulation of E-cadherin, leading to the inhibition of cell migration.

## 4. Methods

### 4.1. Materials and Chemical Reagents

Flaccidoxide-13-acetate was supplied by Dr. Jui-Hsin Su. Rabbit anti-human Akt, FAK, MMP-2, MMP-9, MMP-13, mTOR, PI3K, TIMP-1, TIMP-2, uPA and phosphorylated Akt, mTOR, and PI3K were purchased from Cell Signaling Technology (Danvers, MA, USA). E-cadherin, N-cadherin, lamin A2, and Snail antibodies were supplied by Epitomics (Burlingame, CA, USA). HRP-conjugated goat anti-rabbit immunoglobulin (IgG) was obtained from Millipore (Bellerica, MA, USA). Thiazolyl blue tetrazolium bromide, streptomycin, penicillin, and DMSO were purchased from Sigma (St. Louis, MO, USA). Goat anti-rabbit and horseradish peroxidase-conjugated immunoglobulin IgG and polyvinylidene difluoride (PVDF) membranes were obtained from Millipore (Bellerica, MA, USA). Fetal bovine serum and DMEM were purchased from Biowest (Nuaillé, France).

### 4.2. Cell Culture

HA22T and HepG2 cells were procured from the Food Industry Research and Development Institute (Hsinchu, Taiwan). We used DMEM to culture cells and maintained cells in a 37 °C humidified incubator under 5% CO_2_. The composition of the DMEM culture medium was as follows: 10% fetal bovine serum and streptomycin/penicillin (100 μg/mL and 100 U/mL, respectively).

### 4.3. Cell Viability Assay

Flaccidoxide-13-acetate was dissolved in DMSO. For all in vitro experiments, the final concentration of DMSO was 0.1% *v*/*v*, and the solubility of flaccidoxide-13-acetate in the tissue culture media was good, with no precipitation or turbidity. Cells were first seeded in 24-well culture plates at a density of 1 × 10^4^ per well and treated with flaccidoxide-13-acetate at various concentrations (2–8 μM). A thiazolyl blue tetrazolium bromide (MTT) assay was used to measure the cytotoxicity of flaccidoxide-13-acetate after treatment for 24 h.

### 4.4. Cell Migration Assay

Cell migration was analyzed by seeding HA22T and HepG2 cells (1 × 10^5^ cells per well) in serum-free medium for 24 h, followed by treatment with flaccidoxide-13-acetate at various concentrations (2, 4, and 6 μM). DMSO (0.1% *v*/*v*) as the vehicle control. The migrated cells were fixed to the bottom chamber with 100% methanol and then stained with 5% Giemsa (Merck, Darmstadt, Germany) before counting using a light microscope.

### 4.5. Cell Invasion Assay

Cell invasion analysis was carried out by seeding HA22T and HepG2 cells (1 × 10^5^ cells per well) suspended in serum-free DMEM on the top chamber of Matrigel–coated Transwell inserts. The lower chamber was filled with DMEM containing various concentrations of flaccidoxide-13-acetate (2, 4, and 6 μM). DMSO (0.1% *v*/*v*) as the vehicle control. After 24 h, the Transwell inserts were fixed with 3.7% formaldehyde, followed by Giemsa staining. The invasive cells were counted using a light microscope.

### 4.6. Gelatin Zymography Assay

Activities of MMPs were analyzed by seeding HA22T and HepG2 cells in 24-well culture plates at a density of 1 × 10^5^ cells per well in serum-free medium (200 μL), followed by treatment with flaccidoxide-13-acetate at various concentrations (2, 4, 6, and 8 μM). DMSO (0.1% *v*/*v*) as the vehicle control. To measure the activities of MMPs and uPA generated from HA22T and HepG2 cells, gelatin zymography with 8% gelatin gels was used to analyze the conditioned medium after 24 h. The gel was then shifted to a reaction buffer at 37 °C overnight for enzymatic reaction, followed by staining with Coomassie blue solution and destaining with acetic acid (10% *v*/*v*) and methanol (20% *v*/*v*) solution.

### 4.7. Western Blot Analysis

Following incubation, cells were washed with PBS three times and lysed in 250-μL lysis buffer (5-mM bicine buffer, 4-(2-aminoethyl) benzenesulfonyl fluoride (AEBSF 0.3 mM), leupeptin (10 μg/mL), and aprotonin (2 μg/mL)). Supernatants were obtained after centrifuging (8000 rpm) the cell lysates. Nuclear proteins were extracted to detect Snail using an extraction kit (SK-0001; Signosis, Santa Clara, CA, USA). Cells were collected and rinsed with ice-cold PBS, incubated in working reagent I for 10 min, and then centrifuged (8000 rpm) for 5 min at 4 °C. The supernatant was then collected and stored. The pellet was incubated in working reagent II for 2 h and centrifuged (8000 rpm) for 5 min. The supernatant was then collected for Snail detection. Protein concentrations were determined by the Bradford protein assay (Bio-Rad, Hercules, CA, USA). The total proteins from cell lysates (25 μg) and nuclear proteins (25 μg) were electrophoresed in a 12.5% SDS-PAGE gel. After protein transfer, the PVDF membranes were blocked with 5% dehydrated skim milk and probed with primary antibodies (anti-human Akt, E-cadherin, FAK, MMP-2, MMP-9, N-cadherin, PI3K, Snail, TIMP-1, TIMP-2, mTOR, uPA, p-PI3K, p-Akt, p-mTOR, and actin antibodies) at 4 °C for 16 h, followed by secondary antibodies for 2 h. ECL reagent was used to detect protein expressions. 

### 4.8. Statistical Analysis

Cell viability assay, cell migration, and invasion assay data were collected from three independent experiments and analyzed using Student’s *t*-test (Sigma-Stat 2.0, San Rafael, CA, USA). Results with *p* < 0.05 were considered statistically significant.

## 5. Conclusions

This study revealed that flaccidoxide-13-acetate, a natural product obtained from marine coral, exhibited multiple capacities to suppress bioactivities of HA22T and HepG2 human HCC cells. First, flaccidoxide-13-acetate suppressed the cell migration, cell invasion, activities of MMP-2/-9 and uPA, and activity of upstream molecule uPA. Second, it disrupted the FAK and PI3K/Akt/mTOR signaling pathways. Third, it impaired the EMT process, including the downregulation of Snail and N-cadherin and upregulation of E-cadherin. The hypothetical mechanism of flaccidoxide-13-acetate in HA22T and HepG2 human HCC cells is illustrated in Figure 7. Taken together, these results suggested that flaccidoxide-13-acetate is a good candidate for further development as an anticancer agent to treat human HCC; however, in vivo research is warranted to confirm its effects. 

## Figures and Tables

**Figure 1 marinedrugs-18-00314-f001:**
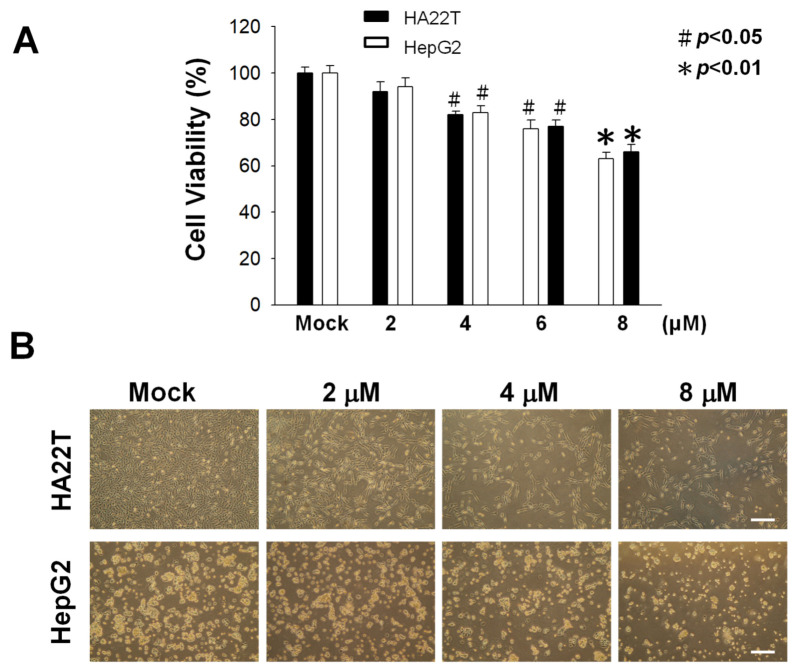
Cell viabilities of HA22T and HepG2 cells treated with flaccidoxide-13-acetate and a control (Mock: DMSO as the vehicle) after 24 h. (**A**) Cytotoxic effects on HA22T and HepG2 cells treated with flaccidoxide-13-acetate exhibiting a dose-dependent manner (# *p* < 0.05 and * *p* < 0.01). (**B**) Morphologies of HA22T and HepG2 cells after 24 h of incubation with 0–8-μM flaccidoxide-13-acetate. Scale bar = 20 μm. The results were obtained from three individual experiments.

**Figure 2 marinedrugs-18-00314-f002:**
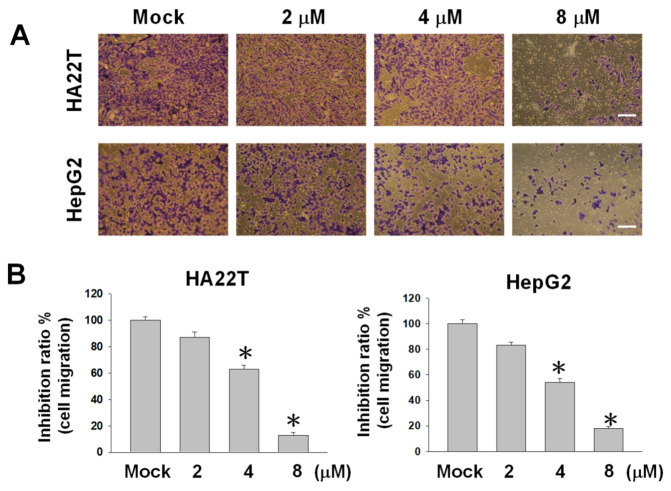
Effects of the flaccidoxide-13-acetate treatment on HA22T and HepG2 cell migrations after treatment for 24 h. (**A**) Images of the migrations of flaccidoxide-13-acetate-treated HA22T and HepG2 cells as compared with controls (Mock: DMSO as the vehicle). Each image was representative of three individual experiments. (**B**) Inhibition ratios of flaccidoxide-13-acetate-treated HA22T and HepG2 cells as compared with controls (* *p* < 0.01). Data were calculated from three individual experiments. Scale bar = 20 μm.

**Figure 3 marinedrugs-18-00314-f003:**
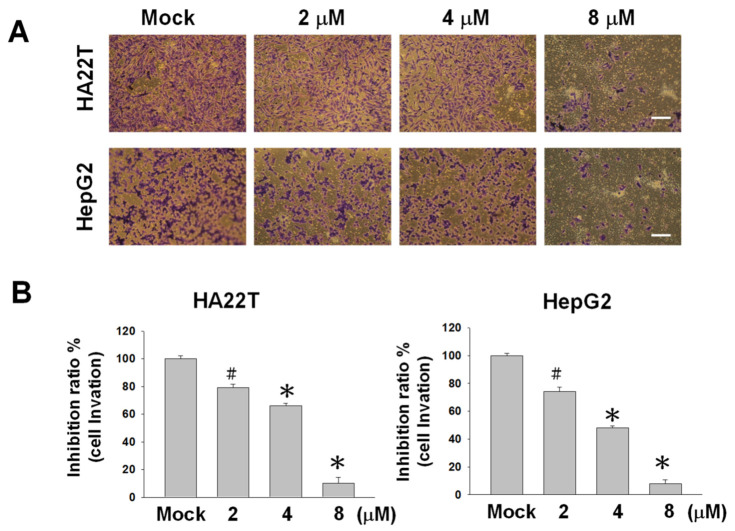
Effects of 24-h flaccidoxide-13-acetate treatments on HA22T and HepG2 cell invasions. (**A**) Images of the invasions of flaccidoxide-13-acetate-treated HA22T and HepG2 cells as compared with the control (Mock: DMSO as the vehicle control). Each image was representative of three individual experiments. (**B**) Inhibition ratios of flaccidoxide-13-acetate-treated HA22T and HepG2 cells as compared with the controls (# *p* < 0.05 and * *p* < 0.01). Data were calculated from three individual experiments. Scale bar = 20 μm.

**Figure 4 marinedrugs-18-00314-f004:**
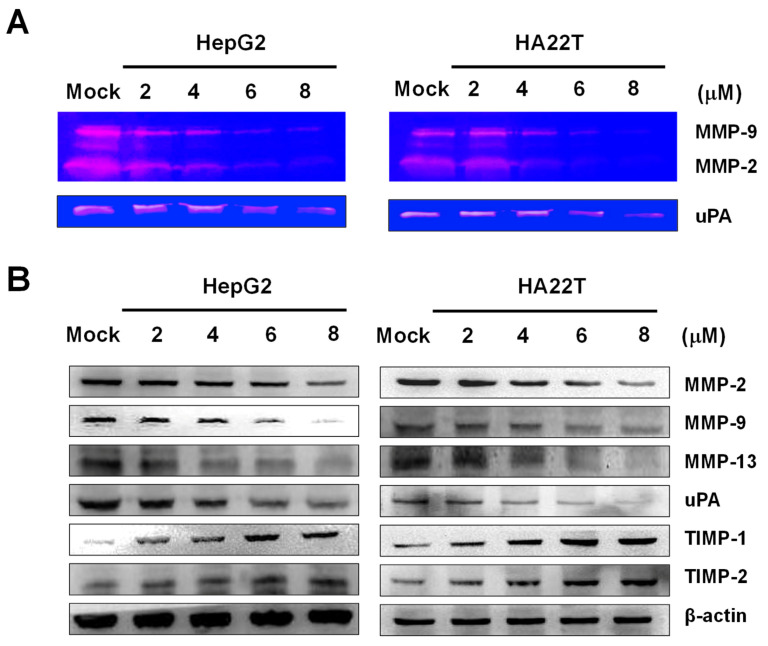
Activities of MMP-2/-9 and uPA and the protein levels of MMP-2, MMP-9, MMP-13, uPA, TIMP-1, and TIMP-2 after treatments of the HA22T and HepG2 cells with flaccidoxide-13-acetate at different concentrations for 24 h. (**A**) Images of the gelatin zymography of MMP-2/-9 and uPA activities. (**B**) Expression levels of MMP-2, MMP-9, MMP-13, uPA, TIMP-1, and TIMP-2. Mock: DMSO as the vehicle control.

**Figure 5 marinedrugs-18-00314-f005:**
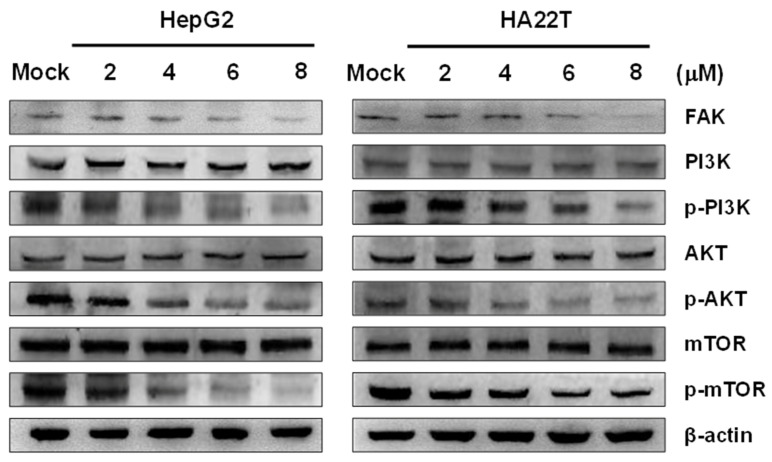
Effects of flaccidoxide-13-acetate on the FAK/PI3K/Akt/mTOR signaling pathway in HA22T and Hep G2 cells. Protein expression levels of FAK, PI3K, Akt, and mTOR and the phosphorylation of PI3K, Akt, and mTOR after treatments with flaccidoxide-13-acetate for 24 h. β-actin: loading control. Mock: DMSO as the vehicle control.

**Figure 6 marinedrugs-18-00314-f006:**
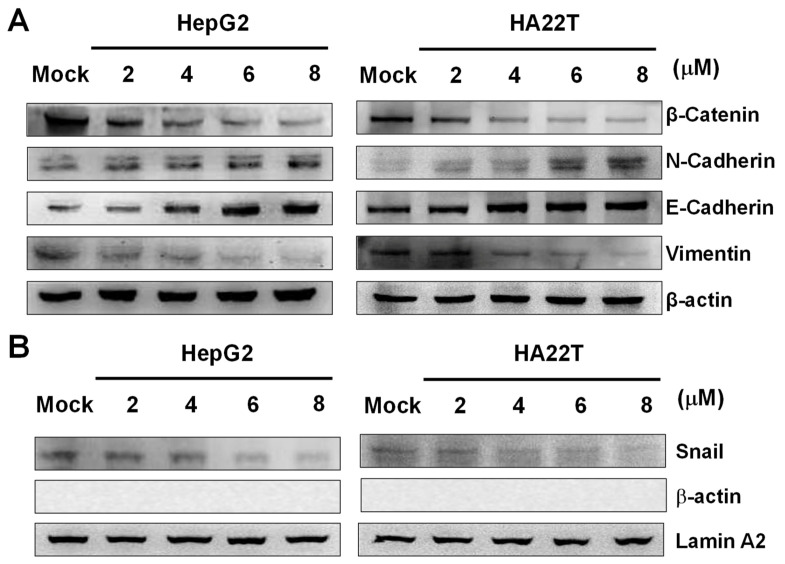
Effects of flaccidoxide-13-acetate at different concentrations on the epithelial-to-mesenchymal transition (EMT) in HA22T and HepG2 cells. (**A**) Protein expression levels of β-catenin, N-cadherin, E-cadherin, and vimentin in cytosol. β-actin: loading control. Mock: DMSO as the vehicle control. (**B**) Protein expression levels of Snail in the nucleus. Internal controls: β-actin for cytosol and lamin A2 for the nucleus.

**Figure 7 marinedrugs-18-00314-f007:**
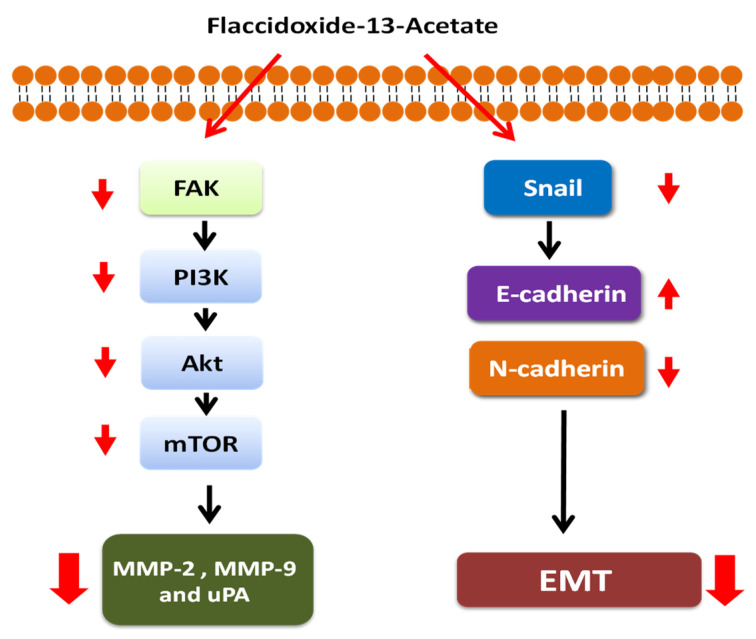
Hypothetical illustration of the flaccidoxide-13-acetate-associated pathway in HA22T and HepG2 human HCC cells.

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
