# Peer review of "Exploring the Mechanism of Flaccidoxide-13-Acetate in Suppressing Cell Metastasis of Hepatocellular Carcinoma"

_marinedrugs, 2020, doi:10.3390/md18060314_

Round 1

Reviewer 1 Report

The authors have assembled a fine manuscript: well-written, with a nice flow to the information, and well organized.

However there is a fair amount of vital information missing from the materials:
4.2: Providing seeding density, and passage number, as well as fetal bovine supplier and lot will facilitate reproducibility.

4.3: The authors do not indicate the well volume, which solvent is used to dissolve flaccidoxide-13-acetate (we are forced to read the figure legends to assume that the solvent is DMSO), the final concentration of DMSO per well is missing (DMSO is well-known to differentiate cell lines at concentrations above 0.1% (v/v) and thus leading to decreased cell viability), a positive control is missing, and the time-course does no correlate to the trans-well assay time of 48 hours.

4.4: Again, volumes, DMSO concentration, and a positive control are necessary.

4.5: My comments are the same here as for Sections 4.4 and 4.3.

4.6: Volumes of conditioned media, how was the conditioned media treated (centrifuged, filtered, stored, concentrated???), DMSO concentrations...etc.

4.7: Once again, the volumes, DMSO concentrations, positive control, and similar are missing in the description of these methods. In addition, a description of the relative centrifugal forces and the centrifuge used is needed.

The lack of these, seemingly small but, vital bits of information severely limit the information that can be reliably gathered from the experiments presented in this manuscript. Leaving this information out also guarantees that the results will not be reproducible - too much information is missing.

While the missing information is critical, a great majority of it should be easily found in the experimenters notebooks, and is, thus, an easy correction.

This well-written and well-organized manuscript will greatly benefit from incorporating the missing information.

Author Response

Reviewer 1

The authors have assembled a fine manuscript: well-written, with a nice flow to the information, and well organized.

However there is a fair amount of vital information missing from the materials:

4.2: Providing seeding density, and passage number, as well as fetal bovine supplier and lot will facilitate reproducibility.

Responds: Thank for reviewer’s suggestion. We have modified the mistake.

4.3: The authors do not indicate the well volume, which solvent is used to dissolve flaccidoxide-13-acetate (we are forced to read the figure legends to assume that the solvent is DMSO), the final concentration of DMSO per well is missing (DMSO is well-known to differentiate cell lines at concentrations above 0.1% (v/v) and thus leading to decreased cell viability), a positive control is missing, and the time-course does no correlate to the trans-well assay time of 48 hours.

Responds: Thank for reviewer’s suggestion. We have modified the mistake.

Effects of flaccidoxide-13-acetate treatment on HA22T and Hep G2 cell migration and invasion after treatment for 24 hours.

4.4: Again, volumes, DMSO concentration, and a positive control are necessary.

Responds: Thank for reviewer’s suggestion. We have modified the mistake.

4.5: My comments are the same here as for Sections 4.4 and 4.3.

Responds: Thank for reviewer’s suggestion. We have modified the mistake and re-corrected them in the experimental method 4.3 and 4.4.

4.6: Volumes of conditioned media, how was the conditioned media treated (centrifuged, filtered, stored, concentrated???), DMSO concentrations...etc.

Responds: Thank for reviewer’s suggestion. We have modified the mistake and added the description in section 4.6

4.7: Once again, the volumes, DMSO concentrations, positive control, and similar are missing in the description of these methods. In addition, a description of the relative centrifugal forces and the centrifuge used is needed.

Responds: Thank for reviewer’s suggestion. We have modified the mistake and added the description in section 4.7

The lack of these, seemingly small but, vital bits of information severely limit the information that can be reliably gathered from the experiments presented in this manuscript. Leaving this information out also guarantees that the results will not be reproducible - too much information is missing.

While the missing information is critical, a great majority of it should be easily found in the experimenters notebooks, and is, thus, an easy correction.

This well-written and well-organized manuscript will greatly benefit from incorporating the missing information.

Responds: Thanks to reviewer for your suggestion. We have corrected these errors and re-corrected them in the experimental method. We will describe these experimental methods more carefully in future papers.

Reviewer 2 Report

This is an interesting manuscript. The experiemnts are well done. I have a few minor comments that will help improve the manuscript to be suitable for publication in this journal.

Figures (Fig. 1B, 2A, and 3A) derived from microscope are not clear at all because of the poor resolution. High resolution images are required.

Fig. 1B should have scale bars and higher magnification images as well. This should be done for all the images in Fig. 2A and 3A.

Figure legends should include the number of replicates (both experimental and biological replicates) as well as the Statistical analysis performed. This should be done for all the figures.

Methods should be sufficiently detailed to properly repeated by other researchers

Author Response

This is an interesting manuscript. The experiemnts are well done. I have a few minor comments that will help improve the manuscript to be suitable for publication in this journal.

Figures (Fig. 1B, 2A, and 3A) derived from microscope are not clear at all because of the poor resolution. High resolution images are required.

Responds: Thank for reviewer’s suggestion. We have modified this error and replaced in Figure 1B, 2A and 3A.

Fig. 1B should have scale bars and higher magnification images as well. This should be done for all the images in Fig. 2A and 3A.

Responds: Thank for reviewer’s suggestion. We have modified this error and replaced in Figure 2A and 3A.

Figure legends should include the number of replicates (both experimental and biological replicates) as well as the Statistical analysis performed. This should be done for all the figures.

Responds: Thank for reviewer’s suggestion. We have modified this error and replaced in all the figures.

 Methods should be sufficiently detailed to properly repeated by other researchers

Responds: Thanks to reviewer for your suggestion. We have corrected these errors and re-corrected them in the experimental method. We will describe these experimental methods more carefully in future papers.

Round 2

Reviewer 1 Report

The authors have addressed the concerns and missing information in the Materials and Methods section.

There remain a few, minor, typos:

Line 22 - flaccidoxide-13-acetate is misspelled

Line 225 - ...purchased from Biowest..

Line 268 - ...the collected for Snail detection.